# Transport Mode and the Value of Accessibility–A Potential Input for Sustainable Investment Analysis

**Jonas Eliasson** [1], **Fredrik Kopsch** [2,3,*], **Svante Mandell** [4] and **Mats Wilhelmsson** [5]

1   Division of Communications and Transport Systems, Linköping University, 58183 Linköping, Sweden; jonas.eliasson@liu.se
2   Division of Real Estate Science, Lund University, 222 00 Lund, Sweden
3   Swedish National Road and Transport Research Institute, 102 15 Stockholm, Sweden
4   National Institute of Economic Research, 120 90 Stockholm, Sweden; Svante.Mandell@konj.se
5   Division of Real Estate Economics, School of Architecture and the Built Environment, Royal Institute of Technology (KTH), 100 44 Stockholm, Sweden; mats.wilhelmsson@abe.kth.se
*   Correspondence: fredrik.kopsch@lth.lu.se

**Abstract:** Accessibility plays an essential role in determining real estate prices and land use. An understanding of how accessibility and changes in accessibility influence real estate prices is key to making decisions regarding investments in infrastructure projects. From an accessibility point of view, it is not clear that there should be differences in valuation depending on the mode of public transport, road, or rail. There are, however, other differences that may affect real estate prices differently. For example, railway stations more often than bus stations, tend to be associated with a higher level of service. In addition, an investment in a railway station may signal a long-term decision from the policymaker. A third possible explanation concerns differences in perceived safety, comfort, and security. This paper aims to study if and how capitalization of accessibility differs between modes of transportation. The findings indicate that rail has a higher impact, both for single-family and multifamily houses. The implication of these findings may be of importance for future infrastructure investments and their corresponding cost-benefit analyses. Incorrect valuations of the benefits of infrastructure investments may result in sub-optimal investments.

**Keywords:** Accessibility; hedonic modeling; public transport; rail; bus

---

## 1. Introduction

Accessibility plays an essential role in determining real estate values, both residential and commercial. The reason is that people need to be able to get from one place to another, from their home to work and schools, from work to grocery stores, and to an array of other locations for the consumption of leisure time. The lower the cost (both in time, comfort, and in monetary terms) for these travels from a particular location, theory predicts, the higher will land values be [1].

Investments in infrastructure can, therefore, play an integral role in reducing friction on real estate markets. By investing in say, a new subway line, policymakers can influence the value of land, and through that, the use of said land. Such investments can likewise be used as economic instruments steering towards a sustainable transport system [2]. In a market with a housing shortage, one response by policymakers can be investments in infrastructure to make previously unbuilt land profitable to build on. In the standard cost-benefit analysis of infrastructure projects, this is generally accounted for through estimates of reduction in travel time. While this may be theoretically correct, it may not cover the full picture. Another approach is to estimate the increased benefits through their impacts on real estate values. If a particular location becomes more accessible, either through reducing travel times to

get to and from that location or reducing other costs associated with accessibility, we expect land values at that location to increase. Accounting for both reductions in travel time and impacts on real estate values is incorrect, as we, in theory, expect these to be the same effects, measured in different quantities.

While using reductions if travel times will result in an identical valuation of infrastructure investments regardless of transport mode, there may be a reason to expect a difference in valuation depending on the infrastructure investment. In particular, it is possible that accessibility by road-based and rail-based public transport capitalizes differently in house prices.

The purpose of the current paper is to analyze whether the capitalization of accessibility differs between transport modes, in particular, road-based (bus) and rail-based (subway). There are a few reasons why there could be differences. For one, investments in rail-based infrastructure provide a signal of commitment. While a bus route can be changed or canceled, with short notice, a train route cannot. Furthermore, there may be differences in how transit stations for road-based and rail-based public transport impact other local amenities. Recently, [3] investigate the impact of subway stations on neighborhood amenities. Among their findings is that subway stations positively impact the number of new openings and increase the diversity of restaurants in the local area.

Moreover, [4] show that rail transit station not only increases the property value due to a reduction in commuting costs but also by attracting retail activity in the area around the station. If house values are positively affected by such amenities, and bus stations are less likely to impact such amenities than subway stations, there is reason to believe that the latter will influence house prices more. Another relevant factor that can explain higher capitalization of accessibility differs between transport modes is that the modes differ in security, safety, and comfort [5].

The current paper adds to the existing literature in two ways: first, our modeling approach is based on a sophisticated measure of accessibility that accounts, not only for distance to a hypothetical city center but for travel times, travel costs, and travel demand (realized and unrealized). A more in-depth description of the accessibility measure is given in Section 3. Second, this paper is to our knowledge the first attempt to simultaneously model accessibility with different modes of public transport. While several studies in the past have been dedicated to studying either the impact of accessibility by rail-or road-based public transport, we attempt to capture both within the same modeling framework to answer the question of how, or if, different modes of public transportation are capitalized differently in housing values.

The remainder of this paper is structured as follows. In Section 2, we briefly discuss the body of literature about accessibility and house prices; the focal point is theoretical frameworks. In Section 3, we present the methodological framework, the accessibility measure, and study design. A presentation of the data used for the analysis is also given in Section 3. The results are presented in Section 4, and Section 5 concludes.

## 2. Literature Review

The most common theoretical point of departure when measuring the capitalization of transportation in housing values is the theory of bid-rents. The theory of bid-rents, first presented by [1], builds on [6] theories of land use. In short, Alonso's bid-rent model for housing markets (or residential bid price curves as Alonso named them) shows a relation between the amount of land (or housing) consumed by the individual, and at what distance from the city this land is located. Given the level of income, the individual is faced with the decision of how much to spend on housing, commuting costs, and all other consumption. Under an even stricter assumption of equal plot sizes, the choice of the individual is reduced to a trade-off between consumption of all other goods and services, and location [7]. A location close to the city center means lower commuting costs, but the higher land price, while a more distant location renders higher commuting costs but less expensive housing. In reality, of course, the individual's choice of where to locate and what quantity of housing to consume depends on a vast, almost inconceivable set of factors and not just distance to a city center. Nevertheless, the theory of bid-rents has been shown to successfully explain land prices as a function of

distance to the city center. One such factor from reality that distorts the linear model is the differences in accessibility at different locations. An apartment (or a house) located close to a railway station will be more accessible, all else equal, than an otherwise identical apartment (or house). The implication is that the theoretically linear rent or price gradients resulting from the bid-rent theory, in reality, will have several "hot spots." Such hot spots may exist concerning good schools [8–10]; good neighborhood composition, e.g., with low crime levels [11–13] or good environmental qualities [14–16]. In short, theories of land value predict that land that is associated with more positively valued attributes (such as good schools, good recreation, proximity to work and leisure activities, etc.) will also be worth more as they will be in greater demand from households. Investments in infrastructure, e.g., construction of a new road or railway station or provision of a new bus route, will change the bid-rent functions. If a certain location receives and investment in infrastructure, e.g., by the provision of a new railway station, the theory predicts that prices at this location will increase due to the increased accessibility to such amenities as the city center, schools, or workplaces.

There are two critical questions with regard to *how* to estimate the value of accessibility. First, the method used to separate accessibility from all other components of the commodity that is housing, and second, how to measure accessibility. Moreover, [17] provides a thorough review and meta-analysis of studies looking at the relationship between railway stations and residential (and commercial) properties. Of the 55 studies (42 residential) included in their analysis, all use hedonic models to estimate the impact of railway stations on real estate values (with differing functional forms).

The hedonic model was first presented by [18], as a way to observe implicit prices. The underlying argument of the hedonic model is that goods can be separated into a finite number of attributes. For housing, such attributes can be the number of bedrooms, quality of the structure, and neighborhood attributes, including transport accessibility. Furthermore, since goods can be separated into several attributes, it follows that the price of a good can be separated into many implicit prices of the individual attributes. The hedonic model has gained much support, particularly within the housing economics literature, as it provides a means to measure implicit prices of such goods that otherwise do not have observable prices on the market. While being the most commonly used, the hedonic model is not the only approach to estimate the effect of accessibility on housing values. For example, [19] uses a difference-in-difference approach to estimate the effect of a new rapid bus transit line in Quebec, Canada.

While the method to estimate its effect, the approach to measure accessibility is subject to more variation in the literature. The most common approach in the literature is the proximity to public transport stations. This is often measured as distance bands (discrete measure) from the station [4,19,20]. Other studies analyze distance to station or a combination of distance (continuous distance) and distance bands. Either Euclidean distances [12], and [21–24], or network-based distances, e.g., walking distance as in [21,22], and [25] have been used in previous studies.

Accessibility, of course, is not merely distance to a transit station. That two otherwise identical houses are equidistant to their respective transit station is not a sufficient criterion for the accessibility for the two to be identical. Such parameters as monetary transport costs, the number of departures, and travel times also influence accessibility. For example, [25] partially accounts for this by including the number of departures at the closest train station. On the other hand, [26] use calculated travel times with bus to account for three different types of accessibility, access to the major city center, shops, and employment centers. Furthermore, a 100-m buffer around bus stations is included to account for the negative externality of being located very close to a bus station. Walking time to the nearest bus station as a measure of accessibility is used in [27] while a discrete measure based on walking time to stations where observed transactions are divided into three categories, within 5 min and between 5 to 10 min walking distance of a station is used in [28]. On the other hand, [29] uses both a dichotomous variable describing the existence of a train station and a variable describing the number of peak trains at the station at the census tract level. In addition, to study accessibility by car, two variables, measuring the commute time to CBD and the average commute time, are used. Moreover, [30] uses a combination

of accessibility measures to evaluate a new subway line. In addition to distance to CBD, several time cost variables are included to account for both travel times and the time it takes to get to the station. To the best of our knowledge, there are no studies that simultaneously account for all dimensions (such as time, distance, cost, travel demand) in their measures of accessibility.

Previous results have found both positive and negative effects of proximity to public transport stations (e.g., [4,23], and [25]). These results are, however, not equivalent to stating that the effects of accessibility are ambiguous, rather, being close to a transit station, be it bus or rail, is associated with positive values in terms of accessibility, but also negative values, in terms of noise [15] and higher crime levels [31]. While some studies are devoted to the effects of a particular station, others are merely interested in the global effect of accessibility. However, there are reasons to believe that the effect of accessibility is not global, but that it can vary over market segments and between stations. In addition to a global regression model, [21] study the effects of different stations in Buffalo, NY. Their findings indicate that proximity to some stations is positively valued, while others are negatively capitalized in housing values. There are several potential reasons for this outcome. One is that some stations are highly associated with other amenities, such as shops, while other stations are more associated with dis-amenities.

On the other hand, the results by [32,33] show only weak evidence that a new suburban transportation station increases retail activity measured as retail employment and no effects in a central location. Another explanation, which is also studied by [25], is that neighborhoods with different socio-economic structures will value accessibility differently. For example, [21] find that the proximity effect is positive for high-income neighborhoods but negative for low-income neighborhoods. Others find that proximity to rail is valued higher in low price segments than high price segments [25]. Moreover, some research results indicate that when the bus alternatives fatality index (a combination of safety, security, and comfort) increases, make the train alternative more attractive and therefore creating a potential higher capitalization into the property prices [5]. In a case study from greater Kuala Lumpur, Malaysia, [34], show that the capitalization premium from light rail transit stations varies substantially in space. Compared to some of the earlier literature, the premium is significant in low-income areas and non-significant in high-income areas. Similar results are found in Tehran, Iran [35]. In a recent paper [36], the results seem to indicate that the positive premium in areas with mixed land uses of access to rail transit based on a case study in Wuhan, China. These results are in accordance with, for example [37], which found that capitalization is higher in the central business district. In the Sydney, Australian, context, [37], find that public transportation under construction provides a positive externality in the surrounding residential area.

Many studies have been devoted to analyzing the effect of either private or public transport on house prices; a few have simultaneously studied both. Studies focusing on the effect of public transport tend to either analyze the effects of road-based (bus) or rail-based (train) public transport. To the best of our knowledge, there has been no attempt to study different modes of public transportation simultaneously. A non-exhaustive summary of previous studies is presented in Table 1. It can be noted that, possibly due to the heterogeneous nature of study areas (within and across countries), no conclusive evidence can be found of a difference in capitalization of different public transport modes in house prices.

**Table 1.** Summary of previous studies.

| Author | Mode of Transportation | Accessibility Measure | Result |
|---|---|---|---|
| Bowes and Ihlanfeldt (2001) [4] | Rail | Discrete distance | Between -18.7% (within a quarter-mile of station) and 3.5% within the 2-3 mile radius of a station. |
| Gibbons and Machin (2005) [12] | Rail | Continuous distance | 9.3% in proximity to new stations. |
| Dubé et al. (2011) [19] | Bus | Discrete distance | Between 2.6% and 6.4% increase for properties receiving a rapid bus transit line. |
| Hess and Almeida (2007) [21] | Rail | Continuous distance | 2% to 5% of median home values if within a quarter-mile of the station, but different for different stations. |
| Cervero and Kang (2011) [22] | Bus | Continuous distance | Land price premiums of 5% to 10% within 300 m of bus rapid transit. |
| Geng et al. (2015) [23] | Rail | Continuous distance | Negative impact on prices within the first 500 m from station, positive and diminishing impact between 500 m and 11 km. |
| Deng et al. (2016) [24] | Bus | Continuous distance | 1.32% to 1.39% per additional 100 m closer to bus rapid transit station. |
| Bohman and Nilsson (2016) [25] | Rail | Continuous distance | Large negative effect of being very close. Low price segments value proximity higher (~1% per meter compared to ~0.4% for high price segment). |
| Mulley (2014) [26] | Bus | Travel time, minutes | 4.2% per extra decrease in minute access time to a motorway, 0.7 to shops, and employment. |
| Rodriguez and Targa (2004) [27] | Bus | Walking time, minutes | Between 6.8% and 9.3% decrease in rent for each additional 5 min walking time to the station. |
| Munuz-Raskin (2010) [28] | Bus | Discrete walking time | 8.7% price premiums for dwelling within 5 min compared to 5 to 10 min walking distance of the station. |

The only attempt to study differences between bus and rail, to our knowledge, is through a meta-analysis [38] that attempts to capture, among other outcomes, land-use impacts of bus and rail public transport systems. The results, concerning land use, are, however, inconclusive.

## 3. Methodological Framework and Data

### 3.1. Methodological Framework

Concerning the methodological framework, we need to establish three things, how to measure accessibility, the method used to estimate the impact of accessibility on house prices, and how to separate the effects of accessibility between different modes of public transportation.

There are several approaches to measuring accessibility in the literature. Perhaps most commonly used is a Euclidean distance measure describing, for instance, distance to the nearest bus station for the individual transaction (see, e.g., [24,25] for two recent applications). Being located closer to a bus station is interpreted as having higher accessibility, all else equal. More sophisticated measures use network analysis to measure the distance by road network or travel time in minutes [26].

In this paper, we make use of measures of generalized transport costs that are calculated for Small Areas for Market Statistics (SAMS) through the transport model SAMPERS (for an in-depth description of the SAMPERS model see [39,40]). The SAMPERS model provides estimates of the logsum, which is a measure of the consumer surplus derived from accessibility (see [41,42]). Since the logsum is based on generalized transport costs, we can extract this particular parameter to express accessibility in a monetary entity. All SAMS areas have their unique generalized transport costs divided for public transport and private transport (by car). Furthermore, we only make use of observations that have occurred in proximity to either a subway station, a bus station, or both. Generalized travel costs can be expressed as (1):

$$GCost_{i,j}^m = \theta_1^m \, travel \, time_{i,j}^m + \theta_2^m \, travel \, cost_{i,j}^m + \ln\left(destinaton \, attractiveness_j\right) \qquad (1)$$

where the generalized transport cost (*GCost)* for mode, *m*, from destination *i* to destination *j*, is the sum of travel time, travel cost, and attractiveness of the destination. $\theta_1^m$ and $\theta_2^m$ ara parameters for travel time and travel costs, respectively. Measuring accessibility in this way provides a more thorough description, as it takes both the cost measured in time and monetary trip expense into account. Moreover, since attractiveness at destination is included, which is based on actual travel patterns, the resulting accessibility measure provides a good representation of what can intuitively be understood as accessibility.

In Section 2, we established that the hedonic pricing model is the, by far, the most common approach to assessing the value of accessibility on housing values. This is also the approach for the current paper. We are primarily interested in estimating the price equation given by (2).

$$price \; = \; \alpha * A_{car} + \beta * A_{subway} + \gamma * A_{bus} + \delta * X + \varepsilon \tag{2}$$

where the price is regressed on three different accessibility measures, car, subway, and bus, as well as a matrix (*X*) if other attributes, location, and dwelling specific. The relation between $\beta$ and $\gamma$ is what is of primary interest in the current paper. In particular, if $\beta > \gamma$ we would be able to conclude that accessibility by subway is valued higher than accessibility by bus, at the same level of accessibility. If instead $\beta < \gamma$ we would, on the contrary, conclude that accessibility by subway is valued less than accessibility by bus. Finally, if $\beta \; = \; \gamma$, we can conclude that the rate of capitalization between transport modes is equal.

Since this study aims to analyze the differences in the capitalization of accessibility depending on the mode of transport, bus, rail, and car, we need a way to differentiate the accessibility measure for public transport between bus and rail. Our measures of generalized transport cost can only differentiate between the two transport modes car and public transport. To differentiate between the two types of public transport included in the accessibility measure, we select our study area to include areas where the bus is the only option for public transport and areas with access to the Stockholm subway line. We include all observations of transactions that have occurred within a 2-km radius of the stations.

The first sample includes bus stations in the municipality of Nacka (The bus alternative includes all stations between Skuru and Hemmesta). These are the observations located in the eastern part of Figure 1. There is no option of public transport by rail along this route. Within the rest of the study area, public transport includes buses, subway, and other commuter trains. To attempt to limit the option as much as possible to only include subway as an option, we exclude all transactions of apartments and single-family houses that fall outside of a 2-km radius of subway stations, the same is done for bus stations (for the subway alternative includes the lines Alvik to Hässelby strand and Liljeholmen to Norsborg.) Observations located close to subway stations are located in the western part of Figure 1. This data set up diminishes the need to control for spatial autocorrelation, as we restrict spatial variation by excluding observations based on their location in relation to each other, and to public transport possibilities.

*3.2. Data*

Two types of data have been gathered for the analysis, transactions of single-family homes, and transactions of cooperative apartments. The transaction data covers the periods 2005 through 2015, while the generalized transport costs are calculated for 2015. The lack of generalized transport costs for earlier periods may be problematic if there is a systematic difference over time in the accessibility between the two groups under study, those with access to the subway line, and those with access to bus only.

Descriptive statistics for the included cooperative apartments are presented in Table 2. In addition to the transaction price of the apartment, the size of the apartment is included measured in square meters and the number of rooms. Cooperative apartments include a monthly fee to the cooperative used to cover operating costs and future renovations. The year the apartment was constructed is

also included, which allows us to control for the age of the apartment. Distance to the closest bus or subway station is calculated using coordinates for each transaction. Only apartments that lie within a 2-km radius of each of the included bus or subway stations are included in the analysis. A few neighborhood-specific variables are included, describing how much of the surrounding area (at SAMS level) is build, how large proportion is high rise and low rice respectively, and how large proportion is single-family houses.

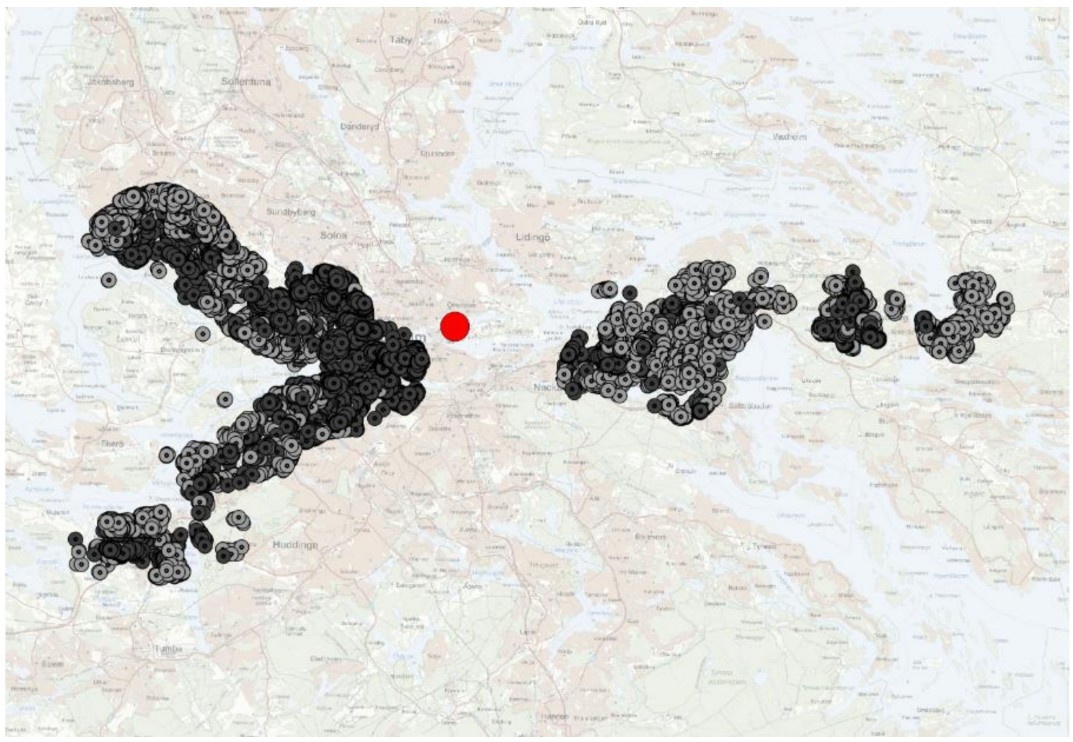

**Figure 1.** Study design and data representation. © Lantmäteriet, Dnr: I2014/00579.

**Table 2.** Descriptive statistics—Cooperative apartments.

| | Total | | Subway | | Bus | |
|---|---|---|---|---|---|---|
| **Variables** | **Average** | **St.dev.** | **Average** | **St.dev.** | **Average** | **St.dev.** |
| Apartment specific variables | | | | | | |
| Price | 1793,136 | 859,129 | 1809,062 | 869,425 | 1725,156 | 810,297 |
| Living area | 67.43 | 22.33 | 66.72 | 21.82 | 70.47 | 24.15 |
| Monthly fee | 3751.39 | 1295.01 | 3691.79 | 1252.44 | 4005.80 | 1435.47 |
| Number of rooms | 2.6 | 1.0 | 2.6 | 1.0 | 2.6 | 1.1 |
| Year built | 1972 | 26 | 1972 | 28 | 1972 | 17 |
| Accessibility measures | | | | | | |
| Distance station | 196.48 | 109.13 | 200.30 | 110.92 | 180.19 | 99.50 |
| Accessibility – Public transport | 111.42 | 3.02 | 111.93 | 2.96 | 109.27 | 2.21 |
| Accessibility - Car | 112.95 | 5.28 | 113.19 | 5.16 | 111.88 | 5.61 |
| Neighborhood specific variables | | | | | | |
| Proportion built area | 0.48 | 0.17 | 0.49 | 0.16 | 0.42 | 0.16 |
| Proportion high rise | 0.63 | 0.33 | 0.66 | 0.32 | 0.54 | 0.34 |
| Proportion low rise | 0.27 | 0.32 | 0.24 | 0.31 | 0.39 | 0.35 |
| Proportion single family | 0.18 | 0.30 | 0.17 | 0.29 | 0.22 | 0.33 |
| Nobs | 22,660 | | 18,359 | | 4301 | |

In total, 22,660 apartments are included in the sample; a vast majority of these, 18,359 is located along with the subway stations, while 4301 are located along with the bus stations described in Figure 1. Light grey dots represent single-family homes, and the darker grey dots represent cooperative

apartments. Stockholm central business district (CBD) is marked in the center or the map with a red dot.

Observations located close to subway stations are located closer to the city center, on average. This can be seen in both Tables 2 and 3 as these observations, for both cooperative apartments and single-family homes exhibit higher values of accessibility.

**Table 3.** Descriptive statistics—Single-family houses.

| | Total | | Subway | | Bus | |
|---|---|---|---|---|---|---|
| **Variables** | **Average** | **St.dev.** | **Average** | **St.dev.** | **Average** | **St.dev.** |
| House specific variables | | | | | | |
| Price | 4223,525 | 2248,949 | 4064,700 | 2252,218 | 4714,003 | 2166,959 |
| Living area | 124.72 | 35.40 | 123.53 | 33.41 | 128.40 | 40.72 |
| Additional area | 37.71 | 36.24 | 40.45 | 35.88 | 29.27 | 36.07 |
| Year built | 1962 | 21 | 1959 | 20 | 1970 | 23.95 |
| Quality points | 28.52 | 3.80 | 28.28 | 3.69 | 29.23 | 4.01 |
| Terrace house | 0.12 | 0.33 | 0.12 | 0.33 | 0.13 | 0.34 |
| Semi detached house | 0.36 | 0.48 | 0.39 | 0.49 | 0.27 | 0.44 |
| Sea | 0.002 | 0.048 | 0.000 | 0.019 | 0.008 | 0.92 |
| Sea view | 0.031 | 0.172 | 0.018 | 0.133 | 0.069 | 0.25 |
| Plot size | 592.34 | 967.57 | 509.21 | 1018.40 | 849.05 | 733.11 |
| Accessibility measures | | | | | | |
| Distance station | 266.74 | 175.39 | 262.69 | 175.52 | 279.24 | 174.43 |
| Accessibility – Public transport | 108.77 | 3.08 | 109.45 | 3.02 | 106.70 | 2.23 |
| Accessibility – Car | 120.40 | 3.80 | 121.48 | 3.37 | 117.09 | 3.06 |
| Neighborhood specific variables | | | | | | |
| Proportion built area | 0.62 | 0.19 | 0.63 | 0.19 | 0.60 | 0.19 |
| Proportion high rise | 0.04 | 0.12 | 0.04 | 0.13 | 0.04 | 0.10 |
| Proportion low rise | 0.92 | 0.15 | 0.93 | 0.15 | 0.88 | 0.16 |
| Proportion single family | 0.88 | 0.19 | 0.89 | 0.17 | 0.83 | 0.24 |
| Nobs | 10,482 | | 7918 | | 2564 | |

It is positive that the two subsamples are similar to each other. The average price 1.8 million Swedish crowns for all transactions, being slightly lower for the sample of apartments with accessibility by bus only. The generalized transport costs differ slightly, with apartments located close to subway stations having slightly higher accessibility. The more substantial differences between the two subsamples can be observed in the neighborhood characteristics. Apartments located along the bus stations are, on average, located in neighborhoods with lower buildings and less densely built areas than apartments located close to subway stations.

Descriptive statistics for the single-family houses are presented in Table 3. For single-family houses, a few more variables are available. In addition to the variables available for apartments, information is also available describing the quality of the property. This is a measure used by the tax agency and is calculated for each house depending on the quality of, for example, materials used in the kitchen. Information is also available describing whether or not the seaside locates the property or if it has a sea view, both of which should affect the value positively. Both the accessibility measures, generalized transport costs, and neighborhood variables are the same for single-family houses for apartments. As for the samples of apartments, single-family houses located by the bus stations have slightly lower values of accessibility than houses located by subway stations.

There is a possibility that the accessibility measures correlate with both apartment/house specific attributes and neighborhood attributes. This would introduce a problem with multicollinearity in our model, which might result in a problem to separate the effects from each other. The large difference between the subsamples is the sample size. While this is unfortunate, it should not affect the estimation of the hedonic model or lead to any systematic errors.

## 4. Results

We estimate two baseline models, one for apartments, and one for single-family houses. The dependent variable in both models is the natural logarithm of the transaction price. The explanatory variables can be divided into four categories, dwelling specific, accessibility, neighborhood-specific, and time-specific. The latter is excluded from the presentation to save space. The results of these two baseline models are presented in Table 4.

**Table 4.** Results—Baseline model.

| Variables | Apartments | | Single-Family Houses | |
|---|---|---|---|---|
| | Coefficient | t-Value | Coefficient | t-Value |
| Constant | 2.0223 | 13.18 | 15.0523 | 47.64 |
| Living area | 0.0143 | 86.14 | 0.0038 | 55.89 |
| Monthly fee | −0.0002 | −92.04 | - | |
| Number of rooms | 0.0875 | 28.98 | - | |
| Year built | 0.0014 | 22.09 | −0.0026 | −20.00 |
| Additional area | - | | 0.0011 | 15.41 |
| Quality points | - | | 0.0035 | 5.52 |
| Terrace house | - | | −0.2104 | −26.42 |
| Semi-detached house | - | | −0.3367 | −54.52 |
| Sea | - | | 0.6484 | 14.55 |
| Sea view | - | | 0.1815 | 14.17 |
| Plot size | - | | −0.0000 | −1.55 |
| Distance station | 0.0002 | 13.69 | 0.0003 | 23.54 |
| Accessibility–Public transport | 0.0713 | 87.78 | 0.0377 | 23.54 |
| Accessibility–Car | 0.0023 | 393 | 0.0018 | 0.86 |
| Accessibility–Public transport* Bus only | −0.0111 | −15.80 | −0.0288 | −10.46 |
| Accessibility – Car * Bus only | 0.0127 | 18.61 | 0.0286 | 11.43 |
| Proportion built area | 0.1499 | 14.58 | 0.1480 | 11.43 |
| Proportion high rise | 0.1740 | 17.35 | −0.0009 | −0.02 |
| Proportion low rise | 0.0707 | 5.98 | 0.0425 | 1.69 |
| Proportion single family | 0.1186 | 9.95 | −0.0779 | −2.86 |
| Nobs | 21993 | | 10443 | |
| Time fixed effects | Yes | | Yes | |
| VIF (average) | 123 | | 141 | |
| $R^2$ | 0.769 | | 0.7878 | |

Note. Dependent variable: natural logarithm of the transaction price. The monthly time fixed effects included in regression but omitted from presentation.

Both models manage to explain roughly 77 percent of the variation in price. All coefficients regarding dwelling specific attributes bare the expected sign, are of reasonable magnitude and are statistically significant at the 5 percent level. The average VIF-values are high, 123, and 141, respectively. There are two main reasons for this. First, we control for temporal fixed effects by adding binary time variables. The in-between correlation of these is, of course, high. Second, and more importantly, the two bus-only interaction terms correlate highly to both each other and to the original accessibility measures. This is a larger problem, the implication of which is that the coefficients, particularly those for accessibility and its interaction terms need to be interpreted with caution.

The coefficient for distance to the station is positive. This might, at first glance, appear counter-intuitive. However, since we have excluded all observations that lie further than 2 km from a station, this result is reasonable. The result does not suggest that proximity to stations is bad per se; it suggests that if you are close to a station, being very close is worse than being located slightly further away.

The interpretation of the coefficients for the accessibility variables is of utmost interest. In general, the results suggest that better accessibility results in higher transaction prices; this is true both for apartments and single-family houses. The results also indicate that the effects of accessibility improvements are larger in areas dominated by subway accessibility compared to areas where the bus is the alternative. The difference in effects is statistically significant, which indicates that the capitalization is different depending on the mode of public transport. The economic interpretation is that for a 1

krona change in transport costs with subway (improvement) prices of apartments can be expected to increase with 0.07 percent. For the average apartment, this means a price increase of 1300 kronor. A corresponding change for accessibility with the bus would change prices with 1100 kronor for the average apartment. The effect is more substantial for single-family houses, where the corresponding changes in prices for a 1 krona improvement in transport costs would mean an increase of 1500 kronor for houses located close to a subway station and 400 for houses close to a bus station.

We also analyze the capitalization of accessibility changes with the car. In general, better accessibility by car has a positive value and leads to higher prices. However, the capitalization effect is much higher where the bus is the only alternative to public transport. A 1-krona improvement of transport costs by car results in 0.002 percent higher prices for apartments located close to subway stations but 0.015 percent for apartments where the bus is the only alternative. The same effects can be seen for single-family houses where the corresponding price changes are 0 percent (for subway) and 0.29 percent (for bus only). These results are intuitive. First, it would be expected that accessibility by car is more critical for single-family houses as they are more likely to live in more periphery areas and are more dependent on the car and own them to a greater extent than households living in apartments. It is also intuitive that accessibility by car is more important where the bus is the only alternative, for roughly the same reasons.

Besides the two baseline models, we have also estimated several separate models for different market segments. In Tables 5 and 6, results are presented for transactions at different distances from Stockholm CBD. In Tables 7 and 8, the models are estimated using only the western parts and the south parts of Stockholm as control groups.

**Table 5.** Results-Apartments.

| | 6–10 Km | | 10–18 Km | |
|---|---|---|---|---|
| **Variables** | **Coefficient** | **t-Value** | **Coefficient** | **t-Value** |
| Distance station | 0.0003 | 17.63 | 0.0000 | 5.11 |
| Accessibility–Public transport | 0.0089 | 4.58 | 0.0727 | 79.34 |
| Accessibility–Car | 0.0151 | 14.49 | −0.0105 | −9.35 |
| Accessibility–Public transport* Bus only | −0.0099 | −9.33 | −0.0231 | −23.86 |
| Accessibility–Car * Bus only | 0.0099 | 9.63 | 0.0237 | 26.54 |
| Nobs | 13,244 | | 8749 | |
| $R^2$ | 0.7593 | | 0.8264 | |

**Table 6.** Results–Single-family houses.

| | 6–10 Km | | 10–18 Km | |
|---|---|---|---|---|
| **Variables** | **Coefficient** | **t-Value** | **Coefficient** | **t-Value** |
| Distance station | 0.0004 | 19.70 | −0.0000 | −0.15 |
| Accessibility–Public transport | −0.0586 | −9.61 | 0.0262 | 13.87 |
| Accessibility–Car | 0.0397 | 9.32 | −0.0154 | −6.45 |
| Accessibility–Public transport* Bus only | 0.0324 | 7.30 | −0.0425 | −13.28 |
| Accessibility–Car * Bus only | −0.0286 | −7.07 | 0.0406 | 13.90 |
| Nobs | 4206 | | 6236 | |
| $R^2$ | 0.7524 | | 0.8190 | |

**Table 7.** Results–Only west parts of Stockholm as control.

| Variables | Apartments | | Single-Family Houses | |
| --- | --- | --- | --- | --- |
| | Coefficient | t-Value | Coefficient | t-Value |
| Distance station | 0.0067 | 10.21 | 0.0003 | 24.77 |
| Accessibility–Public transport | 0.0559 | 41.04 | 0.0268 | 10.88 |
| Accessibility–Car | 0.0067 | 10.21 | 0.0128 | 4.89 |
| Accessibility–Public transport* Bus only | −0.0108 | −15.55 | −0.0515 | −16.41 |
| Accessibility–Car * Bus only | 0.0117 | 17.29 | 0.0486 | 17.16 |
| Nobs | 14513 | | 7989 | |
| $R^2$ | 0.7847 | | 0.7957 | |

**Table 8.** Results–Only south parts of Stockholm as control.

| Variables | Apartments | | Single-Family Houses | |
| --- | --- | --- | --- | --- |
| | Coefficient | t-Value | Coefficient | t-Value |
| Distance station | −0.0000 | −1.26 | 0.0000 | 0.13 |
| Accessibility–Public transport | 0.0786 | 72.17 | 0.0713 | 25.72 |
| Accessibility–Car | −0.0023 | −2.33 | −0.0454 | −15.53 |
| Accessibility–Public transport* Bus only | −0.0229 | −28.92 | −0.0380 | −12.80 |
| Accessibility–Car * Bus only | 0.0249 | 32.28 | 0.0371 | 13.78 |
| Nobs | 11,669 | | 5021 | |
| $R^2$ | 0.8230 | | 0.8226 | |

Without too much focus on the estimated coefficients, a reasonable interpretation of the results is that accessibility with public transport becomes more critical the further from Stockholm CBD you get. The estimates regarding accessibility with public transport are statistically significant. The same results hold for accessibility with the car.

The effect of accessibility on apartment prices where bus is the only option can be differentiated depending on if improvements are made in public transport or by car. Improvements in accessibility by public transport have a lower effect on apartment prices if the bus is the only option. This is true regardless of distance from CBD. Improvements in accessibility by car does; however, increase apartment prices more where bus is the only alternative.

The interpretation is, therefore, that improvements in public transport accessibility had a higher impact if they are made further away from CBD, and more substantial for the subway alternative. This gives further support to our hypothesis that rail transport is more important than road transport. In Table 6, the same model is estimated for single-family houses.

If we analyze the capitalization effects of improved accessibility for single-family houses, we see that there is a difference. First, the results suggest that improved accessibility with public transport is more important the further away from CBD we get. Second, the results for accessibility by car seem to be the opposite. This may very well be due to the problem with multicollinearity discussed previously. Third, accessibility with the car has vastly more important for single-family houses located far away from CBD than accessibility by public transport.

In the baseline model, we compared improvements inaccessibility by public transport where the bus is the only option with a control group that has access to the subway system. The control group includes both dwellings along the west and southbound subway lines. In Tables 7 and 8, we divide these control groups.

When we only use the westbound line of the subway system as a control group, the results are similar to the baseline model. Improved accessibility with public transport has a positive effect on real estate prices, and the effect is larger for both apartments and single-family houses close to the subway stations.

If only transactions along the southbound subway line being used as a control group, the results change slightly. While the coefficients for public transport accessibility remain the same, accessibility with car changes. Accessibility with a car is more important for those areas where public transport is available by bus only.

## 5. Discussion and Conclusions

This paper has aimed to analyze if there are any differences in the capitalization of accessibility depending on the mode of transport. Using generalized transport costs as a measure of accessibility and by designing the data sample to include observations around bus stations as well as subway stations, we can divide accessibility between car and public transport. The separation of accessibility for rail and road-based public transport for the analysis is conducted on one part of Stockholm, where the only option for public transport is bus and two other parts where a subway is an option.

The results from a baseline model indicate that accessibility by public transport is valued higher than accessibility by car. Furthermore, accessibility, both by car and public transport, is capitalized to a greater extent in prices of apartments than single-family houses. However, in the absence of rail-based public transport, car mode will be relatively higher valued, and public transport relatively lower valued. This is the case for both single-family houses as for apartments, although single-family houses see more substantial effects for both.

The results of this analysis provide evidence of a difference in capitalization between rail and road-based public transport. This can be referred to as a rail premium. One limitation of this paper is that we do not aim to say anything regarding the underlying mechanism for the existence of this proposed and observed rail premium. Future research efforts should be aimed at disentangling this question. There are at least three possible hypotheses for why there should be a difference between rail and road-based public transport.

(1) Railway stations act as gravitational zones for commercial activity, which implies that the price premium estimated depends on values at the station rather than the accessibility to other locations it provides. (2) Railway stations signal a long-term commitment in the provision of public transport, which is not signaled by bus routes that are more easily subject to change. (3) As a transport mode, trains are perceived to be safer, more comfortable, and more secure [5].

This leads us to the questions of why this is at all important. This question has to take policymaking into account. From the results that we present in this paper, we can at least distinguish two important policy implications that we have discussed in passing in the introduction to this paper.

(1) As we have previously described, policymakers can use investments in transport infrastructure to influence land use. The connection that we study in this paper is that investments in infrastructure, which increase accessibility, generate higher property values, which in turn may, without mayor supply restrictions, result in changes in land use. Investments in infrastructure can, for example, be used from a policy perspective to steer towards a larger production of housing. The results of this paper suggest that rail-based public transport is preferable as a method of steering land use, seeing that it has a more significant impact on values than road-based public transport.

(2) Investments in infrastructure can also be used to steer the choice of transportation. For example, one of the goals of the policymaker can be to increase public transport from a climate perspective. Public transport is a more environmentally friendly option for transportation. In this paper, we present two pieces of evidence to be used as arguments to this aim. Investments in rail-based public infrastructure are valued higher than road-based dittos. This provides an argument in favor of investments in rail-based infrastructure if the goal is to steer towards public transport. It is simply valued higher. The results also indicate, as mentioned above, that accessibility by car is valued relatively higher in the absence of rail-based public transport. Since the valuation of car-based accessibility is lower when rail-based public transport is an option, we might expect a larger flow to public transport in such cases.

In addition, the indication that rail-based transport and road-based transport are valued differently warrants a further discussion of how the increased accessibilities that they result in should be taken into account in cost-benefit analysis and policy decision making when it comes to infrastructure investments.

**Author Contributions:** Writing this paper has been a joint venture. J.E. was mainly involved in the conceptual design of the study. F.K., S.M. and M.W. wrote an earlier version of the paper. M.W. carried out much of the empirical analysis. F.K. and M.W. have written the finalized version of the paper. All authors have read and agreed to the published version of the manuscript.

**Funding:** This research has benefited from financial support from the Centre for Transport Studies, KTH, Stockholm.

**Acknowledgments:** Comments from attendants at the meetings of the Western Regional Science Association, Santa Fe, 2018 are acknowledged.

**Conflicts of Interest:** The authors declare no conflict of interest.

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
