# Peer review of "Transport Mode and the Value of Accessibility–A Potential Input for Sustainable Investment Analysis"

_sustainability, doi:10.3390/su12052143_

Round 1
Reviewer 1 Report
The aim of the article is to provide statistical evidence of the differences in the capitalisation of accesability according to transport mode.
We would like to highlight a few issues:
1.) The second part of the title is misleading. Article does not contain a "case for sustainable investment"?
2.) Abstract and Introduction (at least the first part) was obviouisly written by a different person than the rest of the paper. It must be corrected (from the language and content point of view). Other part of the paper is much better.
3.) Figure 1 is missing. It should iclude CBD, the two areas and the public transport lines.
4.) The literature should be updated (impact factor of the journal is based on the literature not older than 3 years).
5.) A graphical presentation of the results would be very welcome and would would increase redability of the paper. The area as shuch and data should be better presented.
6.) The conclusions are short and do not present clear added value of the paper.
Author Response
Reviewer 1.
The aim of the article is to provide statistical evidence of the differences in the capitalisation of accesability according to transport mode.
We would like to highlight a few issues:
1.) The second part of the title is misleading. Article does not contain a "case for sustainable investment"?
2.) Abstract and Introduction (at least the first part) was obviouisly written by a different person than the rest of the paper. It must be corrected (from the language and content point of view). Other part of the paper is much better.
3.) Figure 1 is missing. It should iclude CBD, the two areas and the public transport lines.
4.) The literature should be updated (impact factor of the journal is based on the literature not older than 3 years).
5.) A graphical presentation of the results would be very welcome and would would increase redability of the paper. The area as shuch and data should be better presented.
6.) The conclusions are short and do not present clear added value of the paper.
Our Response.
Thank you for your comments.
- Changed to “Transport mode and the value of accessibility – A potential input for sustainable investment analysis.”
- I have language edited the text. Many changes.
- We have included figure 1 in the text, sorry for this. We have also expanded the explanation around figure 1.
- Six new articles are reviewed in the literature section.
- See point 3.
- We have expanded the conclusions section, with particular focus on discussion of the results.
Reviewer 2 Report
Dear editor and dear authors,
Thank you for giving me the exciting opportunity to review the present manuscript for a dynamic journal of sustained academic excellence like Sustainability and thank you for exposing me to an interesting research study respectively.
Research looking at investigating the synergies between land use, transport accessibility, household gains is important when trying to evaluate potential for infrastructure investment. Despite its importance this area of study is rather understudied so this paper is a worthwhile addition to a still lacking literature.
Looking at the paper’s theoretical and empirical merits I am genuinely willing to re-review and eventually accept a refined resubmitted version of the paper. I see this paper positively. There are some issues that should be addressed but the revision task is rather small and feasible.
I will now provide the authors with my critical comments that will guide them on how to improve their article for re-submission. These are listed in the following section, which is written in a more engaging and descriptive tone directed to the authors per se:
- I like that you have set out clearly your research objectives at the end of your introduction. I would perhaps be tempted to have the two explicit goals of your work in a bullet-point form so that they are more visible and obvious to the reader.
- Sections 1.1 and 1.2 need to be separated by the introduction. You need to create a literature review section for 1.1; Section 2 literature review. This is the more natural fit for your paper. Also 1.3 needs to be part of a methodology section. With small adjustments this can be section 3.1 and you present 2. Data can be 3.2 Data.
- The paper lacks a bit in style and presentation when it comes to consistency. In page 6 you have a different font type. Tables 2, 5, 6, 7 are not similar with what feels to be the right table format that you use for Tables 3 and 4. Avoid this brand of mistakes in your revised version
- You also do not employ the numeric referencing style of Sustainability. You have to apply that to be in line with the journal explicit recommendations.
- The conclusions section is interesting but the lack of a discussion section makes it inadequate as the lone section exploring further your results and benchmarking it against the existing literature and against your promised scientific contributions set out in your introduction. I would recommend that you expand this to at least one more page and in one of your new paragraphs you specifically cover the ‘so what?’ factor. Rename it discussion and conclusions. Provide some more policy recommendations that your study might be able to generate. How generalisable are your results to a broader context? Any future studies planned? Note that you have a mistake in the current sub-heading.
Author Response
hese are listed in the following section, which is written in a more engaging and descriptive tone directed to the authors per se:
- I like that you have set out clearly your research objectives at the end of your introduction. I would perhaps be tempted to have the two explicit goals of your work in a bullet-point form so that they are more visible and obvious to the reader.
- Sections 1.1 and 1.2 need to be separated by the introduction. You need to create a literature review section for 1.1; Section 2 literature review. This is the more natural fit for your paper. Also 1.3 needs to be part of a methodology section. With small adjustments this can be section 3.1 and you present 2. Data can be 3.2 Data.
- The paper lacks a bit in style and presentation when it comes to consistency. In page 6 you have a different font type. Tables 2, 5, 6, 7 are not similar with what feels to be the right table format that you use for Tables 3 and 4. Avoid this brand of mistakes in your revised version
- You also do not employ the numeric referencing style of Sustainability. You have to apply that to be in line with the journal explicit recommendations.
- The conclusions section is interesting but the lack of a discussion section makes it inadequate as the lone section exploring further your results and benchmarking it against the existing literature and against your promised scientific contributions set out in your introduction. I would recommend that you expand this to at least one more page and in one of your new paragraphs you specifically cover the ‘so what?’ factor. Rename it discussion and conclusions. Provide some more policy recommendations that your study might be able to generate. How generalisable are your results to a broader context? Any future studies planned? Note that you have a mistake in the current sub-heading.
Our Response
Thanks for your comments!
- We are not sure we understand, however, we hope and trust that the revised introduction will make this issue clear.
- Yes, has been done. Separated section 1. Created a new section 2 with the literature review that has also been extended with some recently published articles.
- Yes, has been done.
- Yes, has been done.
- We have changed the section discussion an conclusins in line with the comments from both reviewers.
Round 2
Reviewer 1 Report
The corrections are accaptable.
There are still some minor typo errors to be checked. For example: Investments in infrastructe can also be used to to steer ...
After this final text editing I propose the publication of the article.
Author Response
We have gone through the text and corrected typos and other errors. Thank you!